# Human Fetal Liver Parenchyma CD71+ Cells Have AIRE and Tissue-Specific Antigen Gene Expression

**DOI:** 10.3390/genes13071278

**Published:** 2022-07-19

**Authors:** Roman Perik-Zavodskii, Olga Perik-Zavodskaya, Yulia Shevchenko, Saleh Alrhmoun, Marina Volynets, Konstantin Zaitsev, Sergey Sennikov

**Affiliations:** 1Laboratory of Molecular Immunology, Federal State Budgetary Scientific Institution Research Institute of Fundamental and Clinical Immunology, 630099 Novosibirsk, Russia; zavodskii.1448@gmail.com (R.P.-Z.); okoneva94@gmail.com (O.P.-Z.); shevcen@ngs.ru (Y.S.); saleh.alrhmoun1@gmail.com (S.A.); mrsmarinavolynets@gmail.com (M.V.); 2Federal State Budgetary Scientific Institution “Siberian Federal Research and Clinical Center of the Federal Medicobiological Agency”, 634009 Tomsk, Russia; limdff@yandex.ru

**Keywords:** CD71+ cells, CD71+ erythroid cells, human fetal liver, AIRE, tissue specific antigens, TSAs

## Abstract

Autoimmune regulator (AIRE) is a multifunctional protein that is capable of inducing tissue-specific antigens’ (TSAs) gene expression, a key event in the induction of self-tolerance, that is usually expressed and functions in the thymus. However, its expression has been detected outside the thymus and cells expressing the gene have been named extra-thymic *AIRE* expressing cells (eTACs). Here, we discuss the finding of *AIRE* and TSAs gene expression in CD71+ cells from human fetal liver parenchyma, which are mostly represented by CD71+ erythroid cells.

## 1. Introduction

Autoimmune regulator (AIRE) is a multi-potent protein [1,2] that plays a major role in the induction of self-tolerance through the expression of tissue specific antigens (TSAs) [3] and their presentation to T-cells by medullary thymic epithelial cells [4] and indirectly by dendritic cells [5] in the thymus. Cortical thymic epithelial cells also express *AIRE* but Nishijima et al. [6] did not observe TSA gene expression in these cells. AIRE-expressing cells were detected outside the thymus and were named extra-thymic *AIRE*-expressing cells (eTACs). Such eTACs were found in secondary murine lymphoid organs [7], where antigen-presenting cells (APCs) [8] and eTACs absence was shown to restrict intrauterine development in mice models [9]. Finding of new eTACs and eTACs in new locations can give us further understanding of the developmental biology and the maintenance of self-tolerance at various stages of ontogenesis at different tissues.

In this work, we analyzed publicly available CD71+ cells’ gene expression data set (GSE199228), found that there is detectable *AIRE* gene expression in human fetal liver CD71+ cells, and decided to validate *AIRE* gene expression as well as check for the induction of selected TSAs (*GCG*, *INS* and *TFF3*) [10] gene expression by *AIRE* by a less noisy pipeline than NanoString, i.e., touchdown PCR, melt curve analysis and DNA gel electrophoresis.

## 2. Materials and Methods

### 2.1. NanoString Data Analysis

We analyzed previously available NanoString data of CD71+ cells from human adult bone marrow and fetal liver parenchyma (GSE199228). We performed background subtraction of the data using the mean of negative controls +2 Standard Deviations in order to filter out noise from the data and considered genes with >2 detected probe count numbers as noise-free.

### 2.2. Study Population

Human fetal liver parenchyma samples (20–22 weeks of pregnancy) were obtained from the “Bank Stvolovih Kletok” LLC (Tomsk, Russia) cell bank; sex is unknown (*n* = 6).

### 2.3. Cell Isolation

We thawed fetal liver parenchyma samples stored in 10% DMSO and 90% FBS (up to 1.5 mL in volume) in a water bath at 37 °C and then washed them with the 6 mL mixture containing 5 mL full RPMI 1640 cell culture medium and 1 mL FBS. We isolated fetal liver parenchyma mononuclear cells using density gradient centrifugation (Ficoll-Paque™ (Sigma-Aldrich, St. Louis, MO, USA) with the density of 1.077 g/mL) at 266 RCF for 30 min.

### 2.4. Cell Sorting

We performed magnetic sorting of fetal liver parenchyma mononuclear cells using a magnetic stand and a magnet (Miltenyi Biotec, 130-042-102, Bergisch Gladbach, Germany) and anti-CD71 MicroBeads (Miltenyi Biotec, 130-046-201) according to the manufacturer’s protocols.

### 2.5. Viability Staining

We measured the magnetically sorted cells’ viability on a Countess 3 Automated Cell Counter (Thermo Fisher Scientific, Waltham, MA, USA) according to the manufacturer’s protocols using Trypan Blue. Trypan Blue staining showed >95% viability of all samples.

### 2.6. Flow Cytometry

We washed magnetically sorted cells in PBS. We used anti-CD71-PE (BioLegend, 334106, San Diego, CA, USA) and anti-CD235a-FITC (BioLegend, 349104) antibody for staining according to the manufacturer’s protocols. Flow cytometry showed >94% purity of the cells. A gating strategy was to isolate singlets from cells, measure CD71 in the singlets and measure CD235a in the CD71+ singlets fetal liver CECs. CD71^+^ erythroid cells (CECs) comprised >90% of the cells (Figure 1).

### 2.7. Total RNA Extraction

We isolated total RNA from cells using the Total RNA Purification Plus Kit (Norgen Biotek, 48400), and measured concentration of the RNA on the NanoDrop 2000c. We froze the total RNA at −80 °C until the reverse transcription of the RNA.

### 2.8. Reverse Transcription

We performed reverse transcription of the total RNA samples (*n* = 6) using RNAscribe RT and oligo-dT primers (Biolabmix, R04-50). We used an input of 100 ng of the RNA. Reverse transcription was conducted as following: 55 °C for 50 min, 80 °C for 10 min.

### 2.9. PCR and Melt Curve Analysis

We performed touchdown PCR with melt curve analysis using UDG HS-qPCR Lo-ROX SYBR (×2) Mix (Biolabmix, MHR033-2040), 1 μL out of the 20 μL of RT product, *AIRE*, *INS*, *GCG* and *TFF3* gene primers at a final concentration of 500 μM. Used primers and predicted amplicon characteristics are presented in the Table 1. All reaction were conducted using 4 technical replicates. Touchdown PCR and melt curve analysis were conducted as following: 55 °C for 2 min; 95 °C for 5 min; 11 cycles of: 95 °C for 20 s, 65 °C -> 55 °C for 30 s (1 °C/cycle decrement), 72 °C for 1 min; 29 cycles of 95 °C for 15 s, 55 °C for 20 s, 72 °C for 30 s; 72 °C for 5 min; and melt curve 95 °C -> 65 °C (1 °C/step decrement). Melt curve plots are shown in Figure 2a–d.

### 2.10. Agarose Gel-Electrophoresis of PCR Products

We took 1 μL of the PCR product for each gene of interest, combined it with 8 μL of nuclease-free water and 1 μL of 10× gel loading buffer. We used a 15 cm wide 2% agarose gel, Bio-Rad power supply and the Bio-Rad gel tray. We ran the gel electrophoresis for 30 min with the power supply set to 100 V (Figure 3).

## 3. Results

Human fetal liver parenchyma CD71+ cells have *AIRE* gene expression.

We studied a previously published CD71+ cells transcriptome dataset and found that there is *AIRE* gene expression in human fetal liver CECs (2/4 samples in the dataset had detectable gene expression) but not in adult bone marrow CD71+ cells (0/4 samples in the dataset had detectable gene expression).

Touchdown PCR validation confirms *AIRE* as well as some tissue-specific antigen gene expression in human fetal liver CECs.

We performed a touchdown PCR with a melt curve and a gel electrophoresis and found *AIRE* gene expression as well as the two out of three selected tissue-specific antigens: *GCG* and *TFF3*, but not *INS* (Table 2).

## 4. Discussion

In this work, we showed AIRE and TSAs gene expression in human fetal liver parenchyma CD71+ cells. CD71+ cells were mostly represented by CD71+ erythroid cells (CECs) both in this and previous works. We propose a hypothesis that human fetal liver parenchyma CD71+ cells had *AIRE* gene expression, which in turn had induced the expression of tissue-specific antigens in these cells. It can also be that CD71+ eTACs help with the maintenance of self-tolerance in the human fetal liver parenchyma by one of the known AIRE-dependent mechanisms, such as generation of T-regs or induction of apoptosis in overzealous T-cells, while leaving any non-TSA-reactive T-cell intact [1]. Taking into consideration the fact that most of the cells in the analyses were CECs, we can also make an assumption that *AIRE* and *TSAs* gene expression took place in CECs. Previous works also showed that CECs can express mRNA and produce proteins from a plethora of cytokines and chemokines [11,12,13,14,15,16], and, therefore, are capable of immunoregulation. Absence of *INS* gene expression can possibly be due to the lack thereof in the current *AIRE*-induced TSA gene expression spectrum, as such events were previously reported [17].

Experimental limitation of this study is the absence of the protein production validation of AIRE, GCG and TFF3 proteins.

## 5. Conclusions

We found *AIRE* and TSAs-expressing cells in an unexpected organ of origin that is the human fetal liver, and which requires further investigation. In the future, we plan to: (1) study *AIRE*’s and TSAs’ transcripts co-localization in a single cell, (2) establish the precise lineage of these CD71^+^ AIRE-expressing cells and (3) show *AIRE*’s and TSAs’ protein production.

## Figures and Tables

**Figure 1 genes-13-01278-f001:**
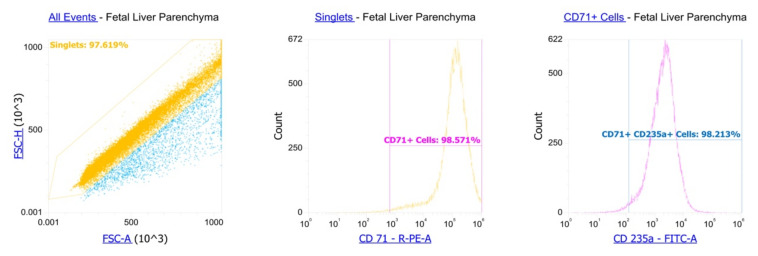
Human fetal liver parenchyma CECs’ purity assessment by flow cytometry.

**Figure 2 genes-13-01278-f002:**
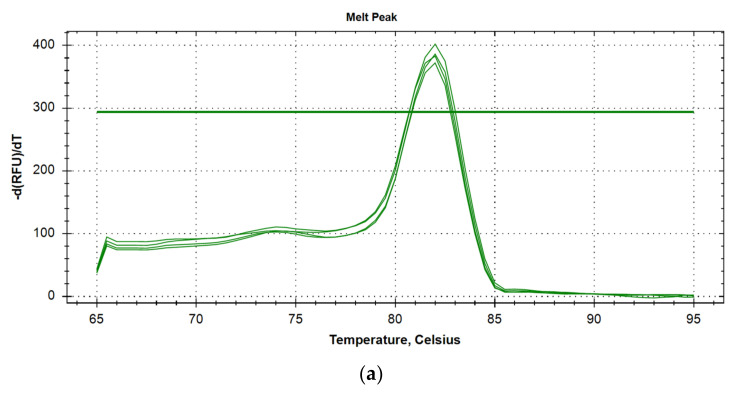
(**a**) Melt curve plot for *AIRE* cDNA amplicons. (**b**) Melt curve plot for *GCG* cDNA amplicons. (**c**) Melt curve plot for *INS* cDNA amplicons. (**d**) Melt curve plot for *TFF3* cDNA amplicons. Melt curve plots for *AIRE*, *GCG*, *INS* and *TFF3* cDNA amplicons.

**Figure 3 genes-13-01278-f003:**
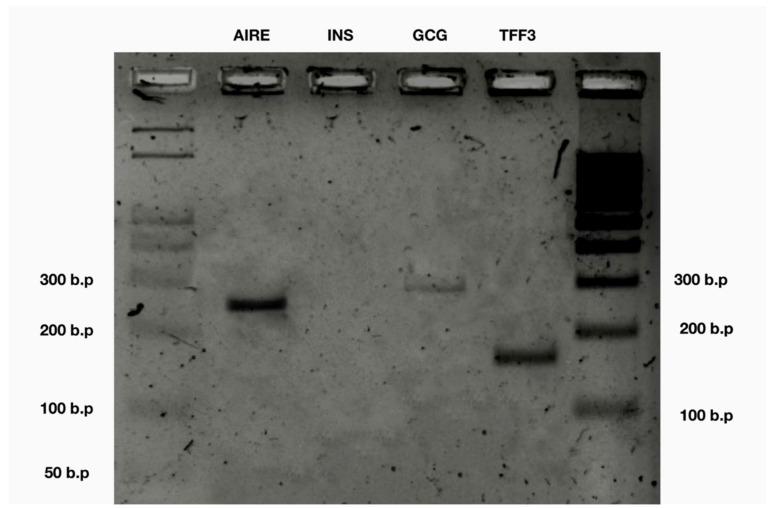
Agarose gel-electrophoresis.

**Table 1 genes-13-01278-t001:** Used primers and predicted amplicon characteristics.

Gene	Forward Primer	Reverse Primer	Amplicon Length	Melting Temperature	Accession
*AIRE*	catctcgaccacttttcagttcag	ccaccatgctgagtaaaataagacag	250 b.p.	82.0 °C	NM_000383.4
*GCG*	ggtgtattctgaggccacattg	tgtggctaccagttcttctattctcc	295 b.p.	73.5 °C	NM_002054.5
*INS*	ggagaactactgcaactagacgcag	ggttcaagggctttattccatctc	89 b.p.	82.5 °C	NM_000207.3
*TFF3*	cacccacgtcacaggaaagc	cgagagtggttgtgaaataaaggac	170 b.p.	78.5 °C	NM_003226.4

**Table 2 genes-13-01278-t002:** Presence of *AIRE* and *TSAs* gene expression in Human Fetal Liver Parenchyma CD71+ cells.

	Human Fetal Liver Parenchyma CD71+ Cells	
Cytokine	Presence	№ of Samples Positive
AIRE	+	6/6
INS	−	0/6
TFF3	+	6/6
GCG	+	6/6

## Data Availability

The previously published publicly available dataset used in this study is available at the Gene Expression Omnibus (GEO) with the accession code GSE199228.

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
