# Peer review of "Human Fetal Liver Parenchyma CD71+ Cells Have AIRE and Tissue-Specific Antigen Gene Expression"

_genes, 2022, doi:10.3390/genes13071278_

Round 1
Reviewer 1 Report
The manuscript by Roman Perik-Zavodskii et al. "Fetal liver parenchyma CD71+ cells have AIRE and Tissue-specific antigen gene expression" is touching on an important topic. Few specific suggestions to improve further include,
i) As a research article, it would be useful to have slightly extended introduction with emphasis on the biological importance of the study and the need for conducting the study.
ii) the results needs to be inferred with biological significance of the findings made in the manuscript. At the moment, it is not reflecting the same.
iii) The genesis of the study is towards taking forward the Nanostring based findings and validating in an independent set of 6 samples using a different method. It is thus important that sensitivity and specificity of the new methods of detection be touched upon. This is especially true as Nanostring can detect upto 2 transcripts. How does that compare with the new method?
iv) It would help to include the time taken for the 2 methods as well as cost for the experiments.
v) And yes, if possible, 6 samples is very lower for significance of the study. Can that be increased?
Best wishes,
Author Response
- i) As a research article, it would be useful to have slightly extended introduction with emphasis on the biological importance of the study and the need for conducting the study.
We added the following: “Finding of new eTACs and eTACs in new locations can give us further understanding of the developmental biology and the maintenance of self-tolerance at various stages of ontogenesis at different tissues.”
- ii) the results needs to be inferred with biological significance of the findings made in the manuscript. At the moment, it is not reflecting the same.
We added the following: “It can also be that CD71+ eTACs help with the maintenance of self-tolerance in the human fetal liver parenchyma by one of the known AIRE-dependant mechanisms, such as generation of T-regs or induction of apoptosis in T-cells.”
iii) The genesis of the study is towards taking forward the Nanostring based findings and validating in an independent set of 6 samples using a different method. It is thus important that sensitivity and specificity of the new methods of detection be touched upon. This is especially true as Nanostring can detect upto 2 transcripts. How does that compare with the new method?
While NanoString can indeed detect as low as 2 copies of a transcript, it cannot distinguish transcripts with low copy numbers from technical. It is mentioned in nSolver 4.0 user manual (nSolver 4.0 User Manual - NanoStringhttps://www.nanostring.com › wp-content › uploads, page 23) that POS_F (Positive control F) is considered below the limit of detection. Median values of POS_F are around 27-30 copies. So, presence of any transcript with detected probe count < 30 in cannot be detected with confidence by any NanoString Sprint platform.
- iv) It would help to include the time taken for the 2 methods as well as cost for the experiments.
While both methods can be used for the same type of experiments (NanoString is less laborious, but way more expensive when compared to qPCR) we just needed to validate the NanoString results by PCR, as it a common practice in the field due to the limit of detection problems discussed above.
- v) And yes, if possible, 6 samples is very lower for significance of the study. Can that be increased?
Sadly, these samples are extremely rare to find due to their fetal origin and we used our full stock of biological replicates in this study. The cell bank from which we acquired the fetal liver parenchyma samples has run out of stock, and no other local cell bank has such a tissue at the moment as well.

Reviewer 2 Report
As the authors state themselves "...further investigation, especially their actual cell lineage and their ability to process and present TSAs in fetal liver, or. maybe elsewhere." The authors present interesting preliminary results, that are however of little significance, if not confirmed by the experiments suggested by the authors themselves.
Author Response
We plan on to validate the generation of the corresponding protein products by western-blot in the future study along with deep full transcriptome single cell RNA sequencing of the CD71+ fetal liver cells. This study is a first step in the direction!

Reviewer 3 Report
In this work, the authors analyzed according to the information available, the expression of the AIRE gene, its expression in CD71+ fetal liver cells, and the induction of the expression of tissue-specific antigens (GCG, INS, and TFF3). This is a very relevant and interesting finding regarding the description of the expression of the AIRE gene in other tissues.
Major comments:
Table 1. Make sure that each nucleotide of the forward and reverse primers are correct, in order to correctly describe the amplicons obtained and that these primers can be replicated in subsequent studies.
Figure 3. It is suggested to repeat the experiment of the visualization of PCR products in a high-resolution electrophoresis gel in 6% polyacrylamide gels derived from a 30% concentration (29 Acrylamide:1 Bis-acrylamide) to improve the resolution of the expected or possible bands secondary hybridization artifacts of the primers and use a 50 bp ladder, in the case of the INS gene the expected product of 89 bp could have been lost from the agarose electrophoretic matrix as it is small, better use polyacrylamide and add an image of this gel.
Due to the fact that several post-transcriptional and translational mechanisms may be regulating the expression of mRNA to protein, it is necessary to validate the generation of its protein products by western-blot, for example, in the case of the presence of the AIRE protein, due to the relevance of this finding in this study.
Minor comments:
Line 28 of the introduction correct wok for work
Indicate whether the simple fetal liver parenchyma was from humans or which organism, although it is obvious and it can be inferred that it is in humans, it must be described in detail in the methodology in order to be replicated in other studies.
Indicate whether the fetal liver parenchyma was simple human or what organism
Describe the density gradient that was used to isolate mononuclear cells from parenchyma (density, Code, and gravities instead of RPMs)
Lines 74, 80-83 indicate that if they are degrees Celsius with the official symbol °C
Briefly include the experimental limitations of your findings in the discussion section.
Author Response
Major comments:
Table 1. Make sure that each nucleotide of the forward and reverse primers are correct, in order to correctly describe the amplicons obtained and that these primers can be replicated in subsequent studies.
We checked primers’ sequences by nucleotide-BLAST, Primer 3 and multiple primer analyzer in silico, we also observed a single peak after a melt curve analysis in vitro, so we can say that these primers match every criterion of quality.
Figure 3. It is suggested to repeat the experiment of the visualization of PCR products in a high-resolution electrophoresis gel in 6% polyacrylamide gels derived from a 30% concentration (29 Acrylamide:1 Bis-acrylamide) to improve the resolution of the expected or possible bands secondary hybridization artifacts of the primers and use a 50 bp ladder, in the case of the INS gene the expected product of 89 bp could have been lost from the agarose electrophoretic matrix as it is small, better use polyacrylamide and add an image of this gel.
We performed 4% agarose gel electrophoresis which had the required resolution (10-500 b.p.). We did not observe any INS corresponding bands, as was expected due to lack of any peak in the melt curve analysis for this gene. We observed other bands at their expected places (Figure can be found in the attached word file).
Due to the fact that several post-transcriptional and translational mechanisms may be regulating the expression of mRNA to protein, it is necessary to validate the generation of its protein products by western-blot, for example, in the case of the presence of the AIRE protein, due to the relevance of this finding in this study.
We plan on doing so in the future study along with deep full transcriptome single cell RNA sequencing of the CD71+ fetal liver cells in order to delineate the origin of eTACs and to test, if AIREand TSAs transcripts are co-localized in a single cell or not. This study is a first step in this direction!
Minor comments:
Line 28 of the introduction correct wok for work
Fixed.
Indicate whether the simple fetal liver parenchyma was from humans or which organism, although it is obvious and it can be inferred that it is in humans, it must be described in detail in the methodology in order to be replicated in other studies.
Indication of species added.
Indicate whether the fetal liver parenchyma was simple human or what organism
Indication of species added.
Describe the density gradient that was used to isolate mononuclear cells from parenchyma (density, Code, and gravities instead of RPMs)
Indication of density gradient added (Ficoll-Paque with the density of 1.077 g/mL).
Lines 74, 80-83 indicate that if they are degrees Celsius with the official symbol °C
Fixed.
Briefly include the experimental limitations of your findings in the discussion section.
Added: Experimental limitation of this study is the absence of protein production validation of AIRE, GCG and TFF3 proteins.

Reviewer 4 Report
It's a well-organized paper, but there are some concerns.
1. Was this study approved by an ethical review committee? Also, did the authors get informed consent for the study from the pregnant women who aborted the fetus?
2. It was unclear how the results of this study would be utilized in the future. I would appreciate if the authors could suggest it.
3. Regarding the temperature, ℃ is written as C, thus correct it.
Author Response
- Was this study approved by an ethical review committee? Also, did the authors get informed consent for the study from the pregnant women who aborted the fetus?
Women were not directly involved in the study. Fetal liver parenchyma samples stored in 10% DMSO and 90% FBS were obtained from the “Bank Stvolovih Kletok” LLC cell bank.
- It was unclear how the results of this study would be utilized in the future. I would appreciate if the authors could suggest it.
While we were only able to detect expression of these genes, we plan on doing a lot more in the future. We plan to study: 1) AIRE and TSAs co-localization in a single cell, 2) establish the lineage of this AIRE-expressing cell, 3) show AIRE’ and TSAs’ protein production. These key questions will be studied by the deep full transcriptome single cell RNA sequencing of CD71+ fetal liver cells (1 and 2) and western-blot (3). This study is a first step in this direction!
- Regarding the temperature, ℃ is written as C, thus correct it.
Fixed.

Round 2
Reviewer 1 Report
Although I would have appreciated more than a line in the revised manuscript submission for the aspects related to the Introduction and result interpretation, but it is ok.
Author Response
We tried our best to elaborate further on the main topic of biological significance that is self-tolerance!
However, even after additional literature study, we could not come up with any additional ideas regarding biological meaning of our discovery.
Reviewer 3 Report
The authors adequately answered the different questions in relation to this preliminary study.
1) I only have the following minor comments: -Describe in the cell isolation section at how many Relative Centrifugal Force (RCF) the samples were centrifuged instead of RPM.
2) Replace figure 3 with the figure that was included as an answer in round 1 about 4% agarose gel electrophoresis
Author Response
1) Changed RPM to RCF
2) Replaced
Reviewer 4 Report
1. P.3, L.96: Change Supportive to Supporting.
2. In the Supporting Figure 2, add a legend to each of a to d.
3. Mention about the Supporting Figure 3 in the text.
Author Response
1) Changed
2) Added
3) Mentioned